# Innovative Fabrication of Hollow Microneedle Arrays Enabling Blood Sampling with a Self-Powered Microfluidic Patch

**DOI:** 10.3390/mi14030615

**Published:** 2023-03-07

**Authors:** Lorenz Van Hileghem, Shashwat Kushwaha, Agnese Piovesan, Pieter Verboven, Bart Nicolaï, Dominiek Reynaerts, Francesco Dal Dosso, Jeroen Lammertyn

**Affiliations:** 1Biosensors Group, Department of Biosystems, KU Leuven, Willem de Croylaan 42, 3001 Leuven, Belgium; 2Institute of Micro- and Nanoscale Integration, KU Leuven, 3001 Leuven, Belgium; 3Manufacturing Processes and Systems, Department of Mechanical Engineering, KU Leuven, Celestijnenlaan 300, 3001 Leuven, Belgium; 4Member of Flanders Make, 3000 Leuven, Belgium; 5Postharvest Group, Department of Biosystems, KU Leuven, Willem de Croylaan 42, 3001 Leuven, Belgium

**Keywords:** hollow microneedle arrays, laser drilling, micro-milling, biofluid sampling, self-powered microfluidics

## Abstract

Microneedles are gaining a lot of attention in the context of sampling cutaneous biofluids such as capillary blood. Their minimal invasiveness and user-friendliness make them a prominent substitute for venous puncture or finger-pricking. Although the latter is suitable for self-sampling, the impracticality of manual handling and the difficulty of obtaining enough qualitative sample is driving the search for better solutions. In this context, hollow microneedle arrays (HMNAs) are particularly interesting for completely integrating sample-to-answer solutions as they create a duct between the skin and the sampling device. However, the fabrication of sharp-tipped HMNAs with a high aspect ratio (AR) is challenging, especially since a length of ≥1500 μm is desired to reach the blood capillaries. In this paper, we first described a novel two-step fabrication protocol for HMNAs in stainless steel by percussion laser drilling and subsequent micro-milling. The HMNAs were then integrated into a self-powered microfluidic sampling patch, containing a capillary pump which was optimized to generate negative pressure differences up to 40.9 ± 1.8 kPa. The sampling patch was validated in vitro, showing the feasibility of sampling 40 μL of liquid. It is anticipated that our proof-of-concept is a starting point for more sophisticated all-in-one biofluid sampling and point-of-care testing systems.

## 1. Introduction

Blood acts as an important window to monitor our body’s health status. Therefore, millions of blood samples are globally collected daily for diagnostic purposes. The invasive character of a venous puncture not only requires healthcare workers with phlebotomy expertise but also holds the risk of needle-stick injuries, accidental infection after non-sterile usage, and denial by people with an aversion to needles [1]. To decentralize the collection of blood, cutaneous capillary blood is a good alternative as it can easily be self-sampled by finger pricking using an external piercing element [2]. However, in addition to still being relatively invasive, obtaining acceptable quantities while maintaining adequate quality is often challenging [3,4,5]. Furthermore, the generated blood droplet has to be deposited manually on, for instance, the inlet of a point-of-care diagnostic test or a dried blood spot card, which often leads to volume losses or inaccurate results.

As a solution for self-sampling, the use of microneedles (MN) received a lot of attention as they are less painful and significantly easier to use [6]. MNs are miniaturized needles that can be employed both to extract biofluids or inject medicine into the skin. For sampling, four major MN types have been described in the literature: solid, porous, hydrogel-forming, and hollow MNs (HMN). When aiming for an integrated blood self-collection device, HMN arrays (HMNAs) are the most interesting as they can form a direct fluidic connection between the cutaneous capillaries and the channels of a microfluidic device [7]. However, they are generally more challenging to fabricate, with many techniques and materials explored so far. Silicon, one of the most common materials used in micro-electromechanical systems, has been processed by means of isotropic etching together with anisotropic deep reactive-ion etching or back-side illumination electrochemical etching [8,9,10,11]. Despite its strength, silicon is brittle and prone to break after insertion in the skin, being problematic as crystalline silicon is not biocompatible [12]. Therefore, polymers such as SU-8, PEGDA, or E-Shell have been used with UV lithographic techniques [13,14,15,16,17,18], whereas PMMA is more suitable to be structured with X-ray lithography or micro-injection molding [19,20,21]. Alternatively, metals have a better strength and ductility [22]. For instance, nickel and titanium have been electroplated or sputtered on structures previously prepared by lithography or etching processes [23], while stainless steel has been used in approaches such as femtosecond-laser machining, electrical discharge machining (EDM) or simply assembling commercially available needles on a base substrate [24,25].

Unfortunately, the above-mentioned HMNA examples are not suitable for blood sampling applications as HMN lengths ≥1500 μm are required to reach the cutaneous capillary network [26,27]. As a creative solution, in-plane configured HMNAs have been fabricated [28]. Though, to be compatible with microfluidic platforms for downstream on-chip sample manipulation, out-of-plane configuration in the form of a patch is required. Hereto, Chaudhri et al. optimized the pre-exposure steps to allow UV lithography on thicker SU-8 layers and fabricated 1540 μm tall HMNAs [26]. However, this technique did not allow for creating a direct fluidic connection with a base plane or beveled sharp tips. Alternatively, Le Thanh et al. presented 1515 μm tall SU-8 HMNAs with pyramidal beveled tips using isotropic etching and inclined UV lithography [29,30]. In addition to SU-8, Miller et al. focused on E-Shell 200 and 300 polymer to create tetrahedron-shaped HMNs by digital light processing 3D printing and 2-photon lithography, respectively [31,32]. Furthermore, metallic HMNAs were achieved as well by a drawing lithography technique and subsequent nickel electro-plating [33,34]. This way, a single out-of-plane hollow MN was achieved with a length of up to 1800 μm. For fabricating a 15° beveled tip, an extra laser cutting step was needed. In follow-up work, the integration of an elastic PDMS chamber with two passive valves allowed the withdrawal of 30 ± 5 μL of blood for downstream processing and analysis on a paper-based sensor for glucose and cholesterol [35].

Among all fabrication procedures for tall HMNAs, the mechanical, biological, and economic advantages of alloys such as stainless steel barely received attention [36]. Furthermore, the main focus on classic microfabrication materials and methods did not solve critical shortcomings such as (1) the challenge to directly create beveled sharp tips, (2) the lack of a direct fluidic connection with the base plane, (3) the need for cleanroom facilities, or (4) the need for intricate fabrication steps that do not allow for batch fabrication. Consequently, the scarcity of useful HMNAs has hampered the progress toward integrated microsampling systems envisioning point-of-care sampling and diagnostic testing.

This work presents the fabrication of stainless steel HMNAs and their integration in a self-powered sampling patch. With the goal of overcoming the challenging material properties of stainless steel, a novel two-step approach by laser drilling and micro-milling is first explored and characterized by brightfield microscopy, X-ray microcomputed tomography (μCT), and pressure drop measurements. Next, the HMNAs are integrated with a self-powered imbibing microfluidic pump by liquid encapsulation (SIMPLE), which is first characterized in terms of pressure generation. Finally, the sampling performance of a SIMPLE-based sampling patch is validated on an in vitro skin equivalent.

## 2. Materials and Methods

### 2.1. Reagents and Materials

Stainless steel grade 304 plates (2 mm thick) and grade 303 rods (50 mm diameter) were acquired from Dejond (Antwerp, Belgium). Milling and chamfer tools were obtained from DIXI Polytool (Le Locle, Switzerland), with a detailed tool list in Appendix A. Acetone was obtained from Acros Organics (Geel, Belgium). PVC plastic films of 180 and 300 μm thickness were bought from Delbo (Maldegem, Belgium). Double-sided pressure-sensitive adhesive (PSA) tapes 200 MP types 7956 MP (153 μm thick) and 7945 MP (127 μm thick) and transfer tape type 467 MP (58 μm thick) were acquired from 3M (Maplewood, MN, USA). Sigma-Aldrich (Overijse, Belgium) delivered Whatman quantitative filter paper grades 40, 43, and 598. Aquapel hydrophobic agent was obtained from Aquapel (Butler County, PA ,USA). Aqueous dye was obtained from Darwin Microfluidics (Paris, France). Agarose powder and glycerol bidistilled 99.5% were obtained from, respectively, Melford (Ipswich, UK) and VWR (Leuven, Belgium).

### 2.2. Hollow Microneedle Fabrication Process

HMNAs were fabricated in a two-step approach, which is schematically shown in Figure 1. First, stainless steel was selected for its (1) mechanical advantage to penetrate skin without buckling or breaking high-AR HMNs, (2) hardness and machinability to obtain sharp and durable tips, (3) availability in medical and hemocompatible grades, and (4) resistance to corrosion keeping the material sterile. On top of these advantages, grade 303 was selected because of its machinability. Then, a batch of workpieces was prepared by cutting the stainless steel rod into 5 mm thick discs by wire-EDM (AgieCharmilles CUT E 600, GF Machining Solutions, Biel, Switzerland) and milling pockets with a resulting thickness of 2 mm (Figure 1a) with a micro-milling center (MMP, Kern Microtechnik, Eschenlohe, Germany). Next, the workpiece with a pocket was manually aligned upside down on the XY-stage of a JK 2000 laser processing system (Lumonics, Rugby, UK) with Nd:YAG medium, operated with a repetition rate of 50 Hz. Percussion laser drilling (Figure 1b), i.e., repeated laser pulses with a pulse duration τ (0.7–1.5 ms), was exploited to create microholes serving as HMN lumens. Combinations of (low) laser energies (1.0 and 1.3 J) and laser pulse durations (0.7–1.5 ms) were screened by measuring the resulting lumen equivalent diameter (ED) and circularity (C) at both the laser entry and exit side by brightfield (BF) microscopy (see Section 2.3). An O_2_ gas flow of 6 bar was used to improve material combustion upon laser-induced heating. Before removing the workpiece from the laser cutter, a reference feature was cut (Figure 1c) to align the workpiece based on the precise location of the lumens with a stereomicroscope (Keyence VHX-500F, Osaka, Japan). The workpiece was then clamped on the micro-milling center, where the needle bevels were cut by an engraving tool (Figure 1d). The bevels were finished first, as longer needle shafts (i.e., with larger length/diameter ratio) result in unstable machining. Afterward, the shafts were finished to achieve final tolerances using an end mill cutter (Figure 1e–f).

The outer shape of the HMNAs around the laser-drilled micro-lumen was designed to yield a length of 1700 μm, an outer diameter of 450 μm and a sharp tip angle of 25°. Because of the geometric limitations of the engraving tool, an inter-needle distance of 650 μm was chosen.

### 2.3. Microscopic and Tomographic Inspections

BF microscopic images were obtained for both sides of the lumen, using an Eclipse Ti-e microscope (Nikon, Tokio, Japan), a pT-100-WHT LED source (CoolLED, Andover, MA, USA), and a Nikon Plan 10× objective. Matlab (MathWorks, Natick, MA, USA) was used to analyze the binarized images to obtain the equivalent diameter (ED) $\sqrt{4A/\pi }$ and circularity (C) $4A\pi /P^{2}$ for each lumen at both sides, with *A* the area and *P* the perimeter.

X-ray micro-computed tomography (μCT) was performed at the KU Leuven XCT Core Facility. Laser-drilled blocks of stainless steel were imaged in a Phoenix Nanotom S (General Electric Research, New York, NY, USA) with a diamond-tungsten target. The source voltage and current were 120 kV and 77 μA, respectively, and the achieved voxel size was 2.2 μm. Furthermore, an aluminum filter of 300 μm was used. This setup was used to acquire 1500 radiographic projections over a 360° angle, using 3 frame averaging. NRecon software (Bruker microCT, Kontich, Belgium) was used for reconstructing the scanned volume and subdividing it into slices. Next, the slices were imported into Matlab for binarization and de-speckling by removing all clusters below 100 pixels. Subsequently, the feature was rotated using the air–metal interface to align the lumen perpendicularly so that the ED and taper angle (to quantify the conical shape of the obtained lumen) could be characterized by function of the depth.

The entire HMNAs were Imaged on a TESCAN UniTOM XL μCT-system (TESCAN XRE, Ghent, Belgium) with a reflection tungsten target. The source voltage and current were 180 kV and 88 μA, respectively, and the achieved voxel size was 4.0 μm. Again, an aluminum filter, here 1 mm thick, was used. A total of 3500 projections were acquired over a 360° angle with an exposure time of 360 ms and two projection averaging. The sample was scanned under a manually set inclination angle to reduce streak artifacts. Volume reconstruction was performed using the TESCAN reconstruction software Acquila (TESCAN XRE, Ghent, Belgium), and image processing, volume rendering, and manual alignment with the principal axes were executed in Avizo 2020.R1 (Visual Sciences Group, Bordeaux, France). The image processing consisted of a median filter with 3 subsequent iterations to reduce the image noise and a thresholding operation to segment the solid phase. The binarized volume was then visualized in 3D using the Volume Rendering function present in Avizo.

### 2.4. Microfluidic Chip Fabrication

The low-cost, rapid prototyping method by Yuen et al. was adopted for the fabrication of microfluidic devices [37]. Chips used for the HMNA flow characterization (Section 3.2) were prepared by cutting microfluidic channels in PSA tape using a Speedy 300 CO_2_ laser cutter (Trotec, Boldon Colliery, UK) and laminated between two PVC layers (180 μm) containing the liquid in- and outlets. The latter features were previously cut out by a Maxx Air 24″ digital craft cutter (KNK, Orlando, FL, USA).

Devices used for the characterization of SIMPLE [38,39,40,41] pressure generation (Section 3.3) were equipped with a hydrophilic stop valve (HSV), hydrophobic barriers (HB), and a filter paper pump (Whatman grade 40) before laminating with the PVC layers (300 μm thick). All paper was cut with a Silhouette Cameo craft cutter (Silhouette, UT, USA). HBs were prepared by cutting 1.5 × 3 mm^2^ rectangular Whatman grade 43 filter paper pieces and by impregnating the filter paper with Aquapel, as previously reported [42]. The HSV consisted of untreated Whatman grade 43 filter paper pieces (same size as HB).

Finally, the sampling patches with integrated HMNA (Section 3.4) were fabricated using transfer tape reinforcement on the paper pump tip. The connection of the finger-press activation spot to the sample liquid channel was provided through an HB so that upon activation, no air could be displaced in the direction of the inlet, and thus the skin, while working liquid displacement toward the filter paper pump would be more effective. PVC layers of 180 μm thick were used and the bottom PVC layer was provided with a 2 mm inlet hole to connect with the HMNA. The latter was attached with a PSA 7945 MP connection ring with inner dimensions of 6 × 6 mm^2^.

### 2.5. Flow Characterization of the HMNAs

Flow characterization of the HMNAs was performed using a LineUp push/pull pressure controller (Fluigent, Paris, France) combined with a flow sensor (Model Medium, Fluigent, Paris, France) to achieve a steady water flow rate while monitoring the required pressure in real-time at 0.1 Hz. A high flow rate of 50 μL/min was pushed through each individual needle lumen. Pushing was chosen over pulling for convenience and to avoid any air between the HMN and the pressure pump system in between different repetitions. This was completed by attaching the fabricated HMNAs to a chip with a single, straight fluidic channel having a width of 1.5 mm and outlet hole diameter of 2 mm and by using a PSA connector piece with a hole < 0.5 mm. Additionally, for each measurement, the attached HMNA was immersed in water to avoid any effect at the water/air interface of droplet formation. Baseline pressures were obtained by repeating for each lumen the same measurement without the needle attached, resulting in pressure drops solely caused by the flow through the HMN lumens. Between experimental repetitions, the HMNAs were cleaned in a Branson 1510 Ultrasonic Cleaner (Branson Ultrasonics, Brookfield, CT, USA) in acetone for 20 min to remove any possible debris. More details on the setup can be found in Appendix A.

### 2.6. SIMPLE Pressure Generation

Measurements of the pressure generated by the SIMPLE were based on Boyle’s law, which describes the relationship between the pressure (P) and volume (V) of an ideal gas given a constant temperature as PV *=* k, with k a certain constant. Therefore, a pre-defined air volume enclosed in an air chamber on-chip was expanded by the SIMPLE. Figure 2 depicts the chip design used. A Harvard PHD 2000 syringe pump (Harvard Apparatus, Holliston, MA, USA) was operated to push the activation liquid (1/50 diluted green aqueous dye in distilled water) further toward the HSV at a flow rate of 35 μL/min, leading to the activation of the pump by pushing the working liquid into the filter paper tip. As soon as the HSV was saturated, the syringe pump remained connected after stopping the flow rate. During the wicking of the working liquid (1/50 diluted blue aqueous dye in distilled water) by the porous material of the SIMPLE, which created negative pressure in the air chamber, the activation liquid was prevented to leak from the HSV to the air plug chamber by means of the HB-2.

The generated pressure was indirectly obtained by measuring the decrease in volume of the working liquid in its chamber. Hereto, the chip operation was recorded by a C920 digital webcam (Logitech, Lausanne, Switzerland) for subsequent analysis using an in-house developed software. The pressure *P_fin_* [Pa] was calculated as:(1)$$P_{fin}=\frac{P_{atm}\cdot V_{in}}{V_{fin}}-P_{atm}$$
with *P_atm_* equal to 101.33 kPa (the atmospheric pressure), *V_in_* (L) the initial air plug volume and *V_fin_* 9 L) the expanded air plug volume. To avoid deflection of the chamber upon pressure reduction, which would lead to an overestimation of the generated pressures, PVC layers 300 μm thick were used. For more details on this setup, the reader is referred to Appendix A.

### 2.7. Validation of a Self-Powered Biofluid Sampling Patch with In Vitro Models

To assess the fluidic performance of the self-powered biofluid sampling patch, three samples were prepared: (i) colored distilled water (1/10 aqueous red dye); (ii) blood mimicking fluid (53.8% *v*/*v*) glycerol diluted in colored distilled water (1/10 aqueous red dye); (iii) agarose gel (2.65% *w*/*v*), prepared by dissolving agarose powder in colored distilled water (1/100 aqueous red dye) on a hotplate (130 °C) under constant stirring with a magnetic stirring bar. When solidified at room temperature, the agarose gel was placed to saturate into a container filled with red-colored distilled water (1/10 aqueous red dye). Before each experiment, its surface was dipped dry with a paper cloth and further dried to air to avoid liquid uptake from the surface. The sampling patch was then gently placed on top of the agarose gel and pushed down perpendicularly to assure insertion of the HMNA shafts without rupturing the hydrogel by horizontal movement.

## 3. Results and Discussion

### 3.1. HMNA Fabrication

The first aim of this work was to fabricate HMNAs with a suitable length to reach the skin’s blood capillaries. Hereto, we chose top-down micromachining of stainless steel. The mechanical advantage of this alloy for skin puncture makes it, consequently, a challenging material with respect to machinability. To create microlumens with a depth of up to 2 mm and an ED between 100–200 μm and to keep the final HMN design sufficiently thin while avoiding too large pressure drops, percussion laser drilling was investigated. Despite the fact that femtosecond lasers are the go-to approach for creating small features, they are not suited for creating high-AR microlumens with this depth due to the typically large taper angles caused by the plasma shielding effect [43]. Therefore, a high-power millisecond laser was exploited aiming to reach the targeted microlumen dimensions with a low amount of taper (<1°) and high C (>0.9).

The overview of the screening results in terms of ED and C is given in Appendix A. As expected, to drill completely through the stainless steel substrate with a success rate of 100%, shorter pulses (0.9 ms) were only needed in case of a higher laser output energy (1.3 J). However, this also resulted in a larger ED, lower C, and overall larger variability. This can be explained by the larger heat-affected zone by a single laser pulse removing more material at once, which can be expected to be less controllable [44]. Hence, only the results for the lowest energy at which the laser setup could be set, 1.0 J, met the requirements, as shown in Figure 3. Figure 3a shows that reducing the output energy to 1.0 J yielded lumens well within the set requirements. Furthermore, Figure 3b demonstrates that for a pulse duration of 1.3 ms, the variability of C is clearly higher compared to larger width. This can be explained by the 8% of the repetitions that failed to completely drill through the material. A pulse duration ≥ 1.4 ms yielded a 100% success rate and lower variability. However, as no significantly different results were obtained between a pulse duration of 1.4 and 1.5 ms, we chose to continue the microlumen fabrication with 1.4 ms in combination with a laser output of 1.0 J.

Even though the targeted microlumens were obtained, the significantly higher ED (*p* < 0.05) for the laser entry versus exit sides suggested a significant amount of taper. To visualize this in detail, microlumens fabricated with the laser settings combination 1.0 J and 1.4 ms were scanned by μCT. The cross-section in Figure 3c shows that, in fact, nearly straight microlumens were achieved with the diameter only considerably increasing at the laser entry side. Based on this, it was decided to laser drill the lumens from the backside of the workpiece to let the wider laser entry side coincide with the HMNA base plate instead of the tip.

For the next fabrication step, the workpiece was aligned on the micro-milling center based on the reference cut to produce 5 × 5 HMNAs according to the procedure described in Section 2.2. The needles were characterized by μCT scanning with an achieved voxel size of 4.0 μm^3^ to inspect the entire HMNAs externally as well as internally. A reconstructed 3D image of a machined HMNA obtained by μCT is shown in Figure 4, demonstrating that none of the 25 HMNs have major artifacts such as broken tips or irregular shapes. Figure 4c portrays the cross-section through one row of the HMNA, showing that the lumens are open after machining, although, a few of the lumens show some irregularities. Their possible impact on the flow through the HMNs is discussed in Section 3.2 (see also Appendix A). Furthermore, the distal tip size was found to be comparable to the image voxel size, indicating sharp tips. Finally, the out-of-plane height and outer diameter were 1647 ± 12 μm and 451 ± 9 μm, respectively. Nevertheless, as a result of the limited scanning resolution, being 7 μm, it should be noted that these measurements are only giving an indication that the aimed requirements for blood sampling are met, without offering a precise characterization.

### 3.2. Flow Characterization of the HMNAs

To assess the machining variability, the pressure drops over the individual needles were analyzed for three different arrays, with three technical repetitions each. A high flow rate (50 μL/min) was applied, based on the maximal flow rates found in the literature for microneedle-based blood sampling [34,45]. Figure 5 depicts the spatial distribution of the average pressure drops per HMN (bar charts with error bars can be found in Appendix A).

The liquid was flowing through all of the HMN lumens and overall, 87% of the measured pressure drops were below 10 kPa and 36% below 1 kPa. Despite the overall satisfying results, some pressure drops were clearly deviating as highlighted in blue for pressure drops > 10 kPa. This variability can be attributed to the variability in the lumen diameter, the presence of burrs or debris, and surface roughness as a result of the microfabrication process. μCT visualizes this partly (see cross-sections with some large burrs in Figure 5 [46]) as it is limited to structures larger than the scanning resolution of 7 μm. In this regard, it is important to stress that there is a strong fourth-order effect from the lumen diameter on the measured pressure drops, according to the Hagen–Poisseuille equation:(2)$$\Delta P=128\frac{\mu LQ}{\pi D^{4}}$$
with ∆*P* (Pa) the pressure drop, μ (Pa·s) the dynamic viscosity, *L* (m) the length of the lumen, *Q* (m^3^/s) the flow rate and *D* (m) the diameter of the lumen. However, it must be noted that Equation (2) assumes smooth and straight channels and can therefore only be used as a mathematical theoretical background and a rough approximation for discussing the flow through the individual microlumens.

To statistically assess the machining variability, the pressure drops for the three arrays were compared by the non-parametric Kruskal–Wallis test. With a *p*-value of 0.093, significant differences between the three HMNAs could not be shown. After removing the statistical outliers, the *p*-value further increased to 0.145. For details on the statistics is referred to Appendix A.

### 3.3. Characterization of SIMPLE Pressure Generation

The microfabricated and characterized HMNAs were then integrated with the SIMPLE microfluidic platform capable of pulling liquids in a self-powered and controlled way [47,48], as a proof-of-concept of a self-powered microsampling patch. An important parameter for cutaneous biofluid extraction from the skin is the pressure difference the system can overcome. The SIMPLE chip design and fabrication were optimized to achieve a maximum generated pressure difference. Figure 6 depicts the pressure difference generated by four chips equipped with filter paper pumps (Whatman 40 quantitative filter paper) shaped as circular sectors of 60°, fabricated as described in Section 2.4. It was shown that the lack of a tight sealing between the filter paper and the surrounding plastics allowed air backflow, leading to the termination of the pumping (see Appendix A). This issue was solved by laminating transfer tape on both sides of the porous paper tip to seal the filter paper pump with the top and bottom PVC layers (see inset of Figure 6). As a result, after one hour, the generated pressure differences by the chips with filter paper pumps reached values of 40.9 ± 1.8 kPa (Figure 6), which is more than sufficient for biofluid extraction from the skin. For comparison, mosquitoes generate pressures up to 7 kPa for capillary blood extraction through their 20 μm diameter fascicle [49,50].

### 3.4. Validation of Self-Powered Biofluid Microsampling Patch with In Vitro Models

Finally, a self-powered microsampling patch was assembled by integrating an HMNA with an SIMPLE chip using a PSA connection. The patch has the size of a regular credit card (86 × 54 mm^2^, design shown in Figure 7a), an ideal feature for self-sampling applications. The chip can be easily activated by a finger press on the activation spot (Figure 7b) to bring the working liquid (blue) into contact with the filter paper pump, which starts creating a pressure difference to withdraw the sample.

Validation of its operational performance was completed for three different in vitro model systems. First, red colored distilled water was drawn from an open liquid container by immersing the HMN tips into the liquid. Next, tests were run with a glycerol-based blood mimicking fluid (Figure 7c) with a viscosity of about 4 cP [51]. Although whole blood is a non-Newtonian fluid, the relatively large lumen and microfluidic channel dimensions are not expected to cause a substantially large Fåhræus–Lindqvist effect, which describes the decreasing apparent viscosity for blood flow through microchannels [52]. However, the surface roughness after laser drilling might result in a further decreasing apparent viscosity, making a Newtonian blood mimicking fluid a worst-case scenario [53]. In a third model setup, the extraction of fluid from the human skin was mimicked by inserting the HMNAs in a 2.65% (*w*/*v*) agarose gel saturated with colored liquids according to [54] (Figure 7d).

Volumes up to 40 μL were successfully sampled with a success rate of 100%. No significant difference in the flow rates was measured for sampling water and blood equivalent (see Appendix A). The average flow rate was reduced, and the variability increased when sampling from skin equivalent. This is expected to be caused by manual HMN insertion or differences in HMNs effectively sampling due to the path of least resistance, e.g., by clogging of the needle lumens with agarose. These results demonstrate that the combination of HMNA and SIMPLE delivers a small and flexible microsampling patch with a large sampling volume capacity.

## 4. Conclusions

This study demonstrated the two-step fabrication and characterization of stainless steel HMNAs and their integration in a self-powered sampling patch toward a decentralized collection of capillary blood. First, percussion laser drilling by pulsed laser drilling on a single spot was explored for different laser settings. Pulses of 1.4 s in combination with a laser output energy of 1.0 J resulted in nearly straight microlumens with ED well within the required range of 100–200 μm and C higher than 0.9. Therefore, these settings were chosen for the subsequent micro-milling of 5 × 5 HMNAs. The μCT scans of full HMNAs showed a good agreement with the designed dimensions. The measured height (base-to-tip) and the outer diameters were on average 1647.1 ± 12.3 μm and 451.1 ± 8.5μm, respectively. Furthermore, all tips appeared to be sharp without artifacts. Flows through each individual lumen were characterized to assess the sampling capacity of each needle in the HMNAs. In particular, for a flow rate as high as 50 μL/min, the average pressure drops were below 10 kPa in 87% of the HMNs. These results show that the proposed two-step protocol is able to produce stainless steel HMNAs with out-of-plane shafts taller than 1500 μm and sharp tips necessary to reach skin capillaries for blood sampling.

Next, the HMNAs were integrated with an SIMPLE microfluidic platform as a proof-of-concept of a self-powered sampling patch. Hereto, it was shown that SIMPLE is able to overcome negative pressure differences up to 40.9 ± 1.8 kPa. Finally, sampling patches with an integrated HMNA were validated by sampling up to 40 μL using three in vitro models: water, a blood-mimicking glycerol solution, and a skin-mimicking agarose gel.

This study showed by a proof-of-concept that a flexible sampling patch with large capacity can be obtained by integrating an HMNA with an SIMPLE pumping unit. A unique advantage of the SIMPLE technology is that it allows for further manipulation of the withdrawn sample in a downstream microfluidic network, e.g., precise metering, loading of dried blood spots, or a myriad of diagnostic applications completely on-chip [41,48,55]. Furthermore, the fabrication of tall HMNs in stainless steel has been proven to be possible and improvements to minimize the production anomalies resulting from machining this challenging material have been suggested by the authors. Finally, the possibility to make this chip flexible is interesting for interacting with highly variable curvature of different bodies and body parts in a “wearables” format during sample acquisition.

To conclude, follow-up work will focus on the interaction with biological materials to assess the penetration efficiency, the occurrence of hemolysis, and the risk of blocking the microlumens as a result of blood clot formation, protein adsorption, or tissue debris.

## Figures and Tables

**Figure 1 micromachines-14-00615-f001:**
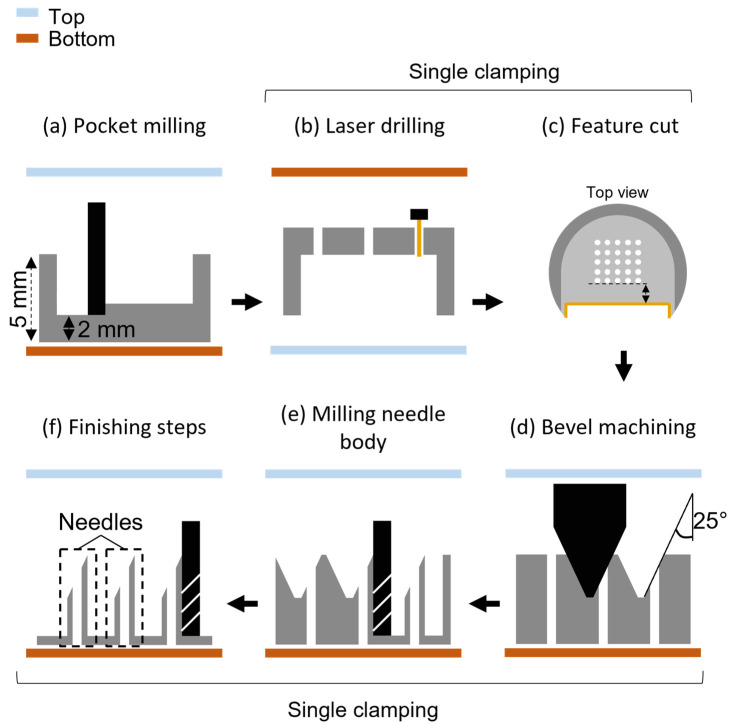
Overview of the HMNA fabrication process. (**a**) A pocket with a 2 mm thick bottom is created, in which (**b**) lumens are laser ablated, followed by (**c**) a reference feature cut (yellow line) in a single step. Then, in a single clamping on the micro-milling center, (**d**) the bevels were machined first followed by (**e**) milling the needle body and (**f**) finishing steps.

**Figure 2 micromachines-14-00615-f002:**
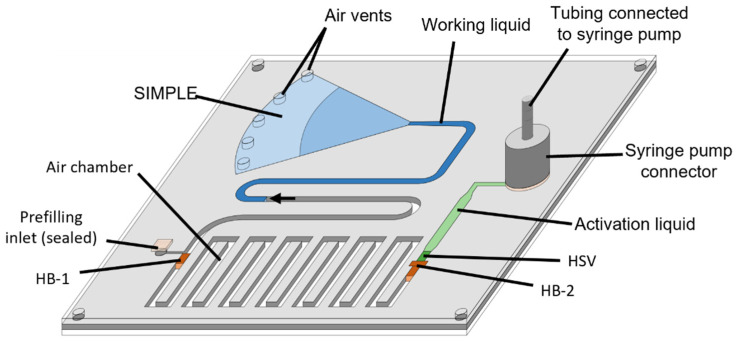
Chip design of a working chip as used for the SIMPLE pressure generation measurements. A syringe pump was used to push the activation liquid (green) towards the hydrophilic stop valve (HSV), which activated the SIMPLE operation by pushing the working liquid (blue) into the filter paper (indicated by the black arrow at the receding liquid front). By effect, the air in between the hydrophobic barriers HB-1 and HB-2 (orange) expanded as quantified by following up the working liquid volume.

**Figure 3 micromachines-14-00615-f003:**
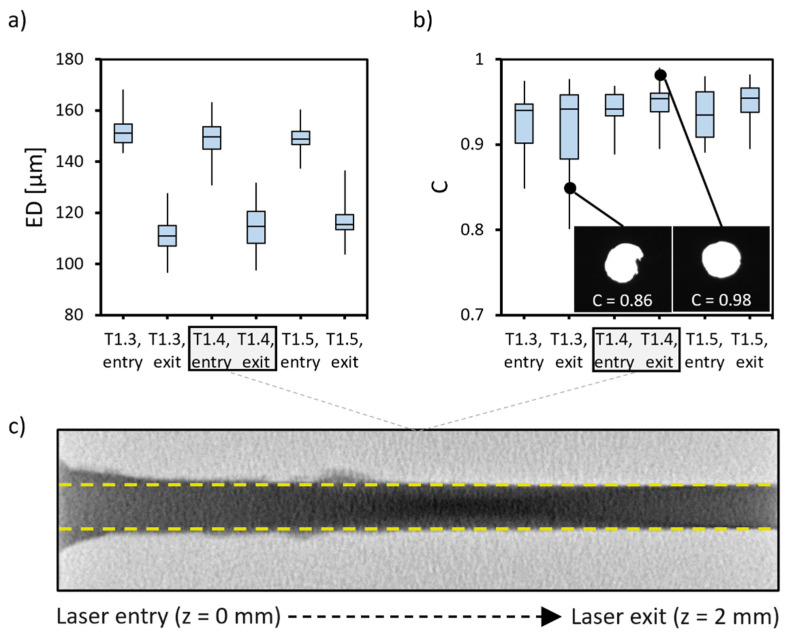
Boxplots for a laser output energy of 1.0 J showing (**a**) the obtained EDs of laser drilled lumens at the laser entry and exit sides for different pulse durations (1.3, 1.4, and 1.5 ms), and (**b**) the C values for the same laser drilled lumens. The boxes in the boxplots display (from top to bottom) the upper quartile, the median and the lower quartile. Whiskers indicate the minima and maxima (*n* > 22 for all conditions) and insets show examples of BF microscopy images obtained and the corresponding results. (**c**) A cross-section by μCT scanning of a lumen percussion laser drilled with energy 1.0 J and pulse duration 1.4 ms. The yellow dashed lines indicate the width at the laser exit side, demonstrating the nearly straight microlumen and substantial widening near the laser entry side.

**Figure 4 micromachines-14-00615-f004:**
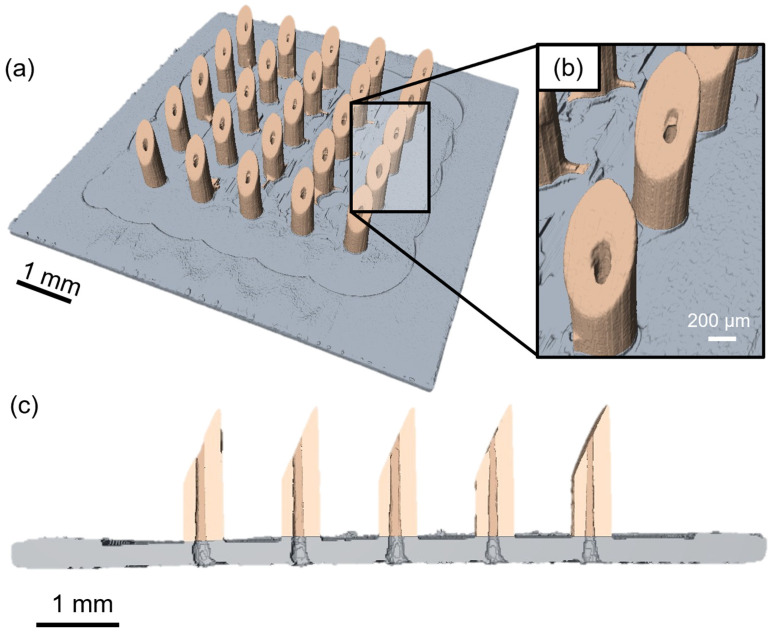
(**a**) Three-dimensional reconstruction of a full 5 × 5 HMNA with indicative lengths of 1647 ± 12 μm and outer diameters of 451 ± 9 μm. False colors (grey and orange) were added to increase the visibility of the machined features. (**b**) Close-up view of a 2 MN shafts. (**c**) Cross-sectional view of the same HMNA showing 5 open lumens and sharp tips as a result of the chosen bevel angle of 25°.

**Figure 5 micromachines-14-00615-f005:**
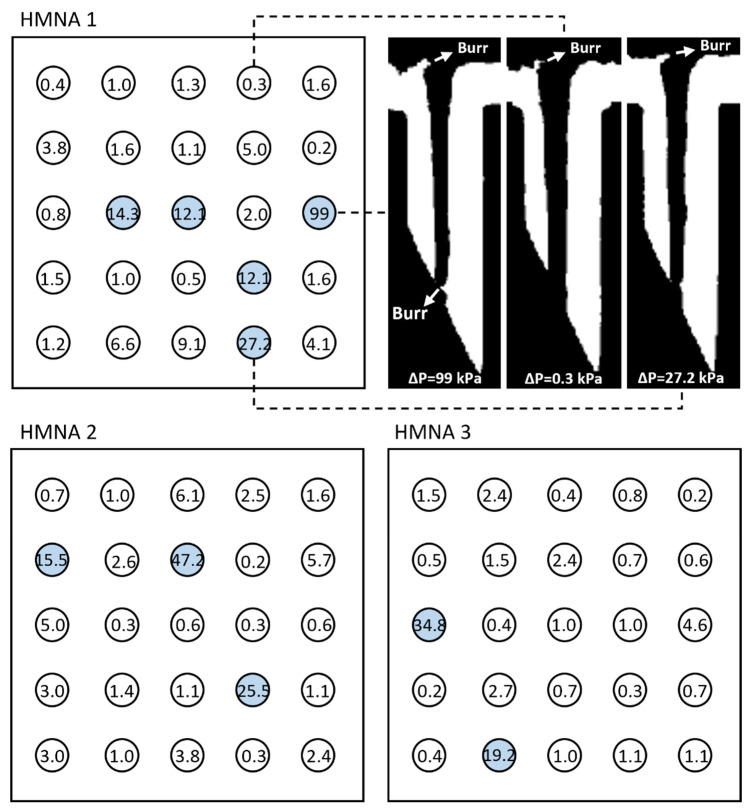
Spatial distribution of the measured average (*n* = 3) pressure drops (kPa) for 3 different arrays at the flow rate of 50 μL/min. Pressure drops > 10 kPa are indicated in blue. For information on the errors resulting from the three technical repetitions, the reader is referred to Appendix A. The presence of machining artifacts such as burrs can partly be visualized by μCT, as shown by the binarized cross-sections of 3 HMNs for HMNA 1.

**Figure 6 micromachines-14-00615-f006:**
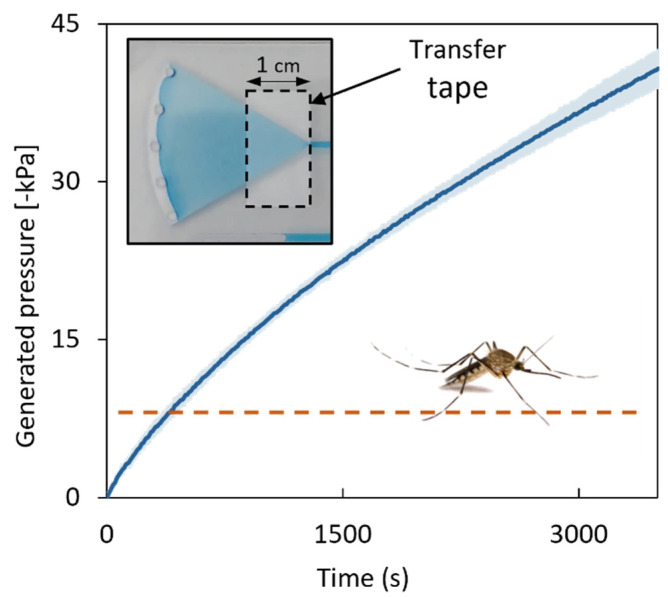
Average generated pressures by the SIMPLE chip reinforced by sealing the filter paper pump tip with transfer tape (see inset). The shaded area represents 1 standard deviation (*n* = 4). The orange dashed line depicts the reference pressure difference as generated by a mosquito when pulling blood from skin capillaries.

**Figure 7 micromachines-14-00615-f007:**
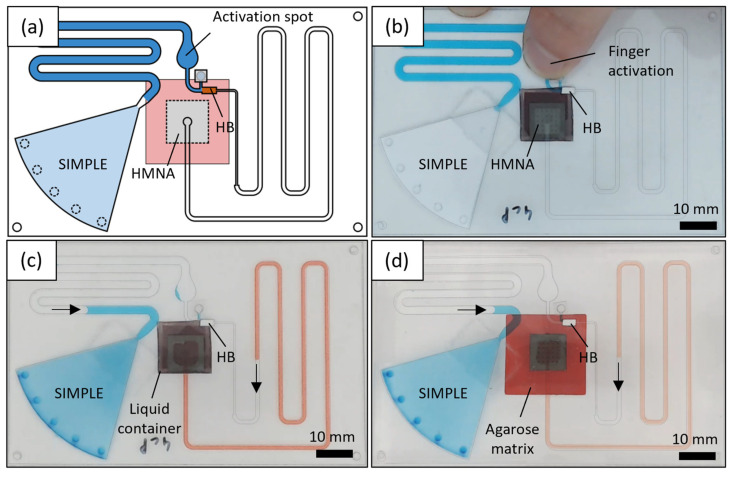
(**a**) Design of the sampling patch used in this experiment. (**b**) Snapshot of a sampling patch during finger press activation. (**c**) Sampling blood-mimicking fluid from an open liquid container. Arrows indicate the direction of the fluid flow. (**d**) Sampling patch while sampling from agarose skin-mimicking matrix. Arrows indicate again the direction of the fluid flow.

## Data Availability

Data available on request.

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
