# Peer review of "Innovative Fabrication of Hollow Microneedle Arrays Enabling Blood Sampling with a Self-Powered Microfluidic Patch"

_micromachines, 2023, doi:10.3390/mi14030615_

Round 1

Reviewer 1 Report

The authors attempt to develop hollow MNs and use them for self-powered micro-sampling. 

The paper would be of interest but I recommend a couple of corrections as follows:

I see that the authors have used H-P equation to workout the flow through the channel. This obviously means that the authors assume that the equation is valid for the cases studied here. Just to ensure that this all reflects on the assumptions of HP equation can the authors discuss how they confirm that the internal surface of the HMN is perfectly smooth.

There is a good amount of discussion on the ID of the HMN - how significant is the variations observed here to state the HP equation is valid?

I could not see any graphs as such for flow and pressure drop for the micro-sampler - could this be provided? 

I would also like to ask the authors to clarify a couple of points in the paper: 

1. What was the reason behind choosing stainless steel for the MN? Could a 3D printed HMN be easier/difficult to fabricate?

2. There is no discussion on the fouling or blockage of the HMN as far as I can see. Could this be addressed/discussed in the paper if not already. 

3. What is the insertion behaviour when/if they are  inserted into skin tissue? any work done in this regard?

Reviewer 2 Report

The author describes a fabrication approach of stainless steel hollow microneedle arrays (HMNAs) integrated with a self-powered sampling patch. The manuscript is well organized. The study is well investigated and presented. It can be accepted.

Author Response

The authors would like to thank the reviewer for taking the time to review this manuscript and for the positive feedback.

Reviewer 3 Report

The authors report a method to fabricate hollow microneedle arrays for blood collection. It is clearly illustrated why developing such a tool is necessary and challenging. The authors thouroughly described the steps of fabrication and the characterization of the microneedles. The feasiblity of operating the device to meter liquid in was demonstrated with water, which should be replaced by blood or at least artificial blood that mimics the viscosity and the compositioin of a blood sample. In addition, 

1. the volume sampled in was low, what is the maxiumum voulme of this device can handle with? and what is the sampling time? A volume-to-time chart might be helpful.

2. What is the limitation of the needle diameter? Can they be milled thinner to further reduce painfulnes? 

3. It seems that the sampling process was inhibited by the pressure drop of the channel and air plug, which will largely preclude the use of this device without employing an external pump. Please illustrate how to address this issue. 

Round 2

Reviewer 1 Report

The authors have addressed my comments and questions and I am happy to recommend an acceptance of the paper.